# Cloud Optimized Raster Encoding (CORE): A Web-Native Streamable Format for Large Environmental Time Series

**Ionuț Iosifescu Enescu [1,*][iD], Lucia de Espona [1][iD], Dominik Haas-Artho [1], Rebecca Kurup Buchholz [1], David Hanimann [1], Marius Rüetschi [1], Dirk Nikolaus Karger [1], Gian-Kasper Plattner [1], Martin Hägeli [1], Christian Ginzler [1][iD], Niklaus E. Zimmermann [1][iD] and Loïc Pellissier [1,2]**

[1] Swiss Federal Institute for Forest, Snow and Landscape Research (WSL), CH-8903 Birmensdorf, Switzerland; lucia.espona@wsl.ch (L.d.E.); dominik.haas@wsl.ch (D.H.-A.); rebecca.kurup@wsl.ch (R.K.B.); david.hanimann@wsl.ch (D.H.); marius.rueetschi@wsl.ch (M.R.); dirk.karger@wsl.ch (D.N.K.); gian-kasper.plattner@wsl.ch (G.-K.P.); martin.haegeli@wsl.ch (M.H.); christian.ginzler@wsl.ch (C.G.); niklaus.zimmermann@wsl.ch (N.E.Z.); loic.pellissier@wsl.ch (L.P.)

[2] Landscape Ecology, Institute of Terrestrial Ecosystems, ETH Zurich, CH-8092 Zurich, Switzerland

\* Correspondence: ionut.iosifescu@wsl.ch

**Abstract:** The Environmental Data Portal EnviDat aims to fuse data publication repository functionalities with next-generation web-based environmental geospatial information systems (web-EGIS) and Earth Observation (EO) data cube functionalities. User requirements related to mapping and visualization represent a major challenge for current environmental data portals. The new Cloud Optimized Raster Encoding (CORE) format enables an efficient storage and management of gridded data by applying video encoding algorithms. Inspired by the cloud optimized GeoTIFF (COG) format, the design of CORE is based on the same principles that enable efficient workflows on the cloud, addressing web-EGIS visualization challenges for large environmental time series in geosciences. CORE is a web-native streamable format that can compactly contain raster imagery as a data hypercube. It enables simultaneous exchange, preservation, and fast visualization of time series raster data in environmental repositories. The CORE format specifications are open source and can be used by other platforms to manage and visualize large environmental time series.

**Keywords:** very large geodata; raster time series; environmental open data; open software; web-EGIS; data cube; hypercube; COG; CORE; EnviDat

## 1. Introduction

Image data cubes are a new paradigm for providing access to large spatio-temporal Earth Observation (EO) data [1]. They represent a generic concept used to organize data with multiple dimensions, with EO imagery data cubes typically having three dimensions: latitude, longitude, and time [2]. This paradigm is being used in various national and global EO data repositories, such as the Swiss Data Cube [3], the Earth Observation Data Cube [4], the Earth System Data Cube [5], and Google Earth Engine [6]. Such EO repositories recognized the need for enabling data access with web-based workflows [7].

The disruptive changes brought by web-based workflows imply that data cubes also need to evolve in how they are physically stored, compressed, and made available using cloud-native access patterns, based on the hypertext transfer protocol (HTTP) capabilities. This evolution can already be observed in the recent development of the Cloud Optimized GeoTIFF (COG), an "imagery format for cloud-native geospatial processing" [8]. COG is a capable format for web-optimized access to raster data, having a particular internal organization of pixels that enables clients to request portions of the file by issuing HTTP range requests. The COG internal organization is based on tiles (covering square areas from the main raster image) while compression is being used for a "more efficient passage online" [8]. The COG state-of-the-art format is applicable to single raster images.

It remains challenging to make available large environmental geodata sets in an efficient manner, especially when they are in the form of time series. For example, scientists at the Swiss Federal Research Institute WSL aspire to provide open and easy access to modified Copernicus data for Switzerland (www.copernicus.eu, accessed on 10 August 2021), with a special interest in contemporary high-resolution data of the Sentinel missions [9]. Cloud-cleared composite reflectance or index products provided as analysis ready data (ARD) will relieve users from costly preprocessing steps [10,11]. The data volume produced by the Sentinel missions is quite significant, with one single compressed (.zip) tile having around 1 GB. This is due to the fact that the Multi Spectral Instrument (MSI) of the Sentinel-2 (S2) mission produces high spatial resolution data (from 10 m to 60 m depending on the spectral band) with a total of 13 bands covering the visible to the short-wave infrared [12]. Similarly, climate model outputs represent other instances of large data volumes being published by WSL scientists. Examples for climate model outputs are the "High resolution climate data for Europe" [13] and the latest versions of the "Climatologies at high resolution for the Earth's land surface areas—CHELSA [14]". These data sets contain climate model data such as temperature and precipitation patterns at high resolution for various time periods, in the form of gridded time series with thousands of layers for a single climate variable.

Environmental data publishing repositories need to undertake a substantial transformation towards managing and visualizing large datasets with web-based workflows. EnviDat (accessible at www.envidat.ch, accessed on 10 August 2021) is the environmental data portal of the Swiss Federal Research Institute WSL providing unified and managed access to environmental monitoring and research data. EnviDat actively promotes good practices for Open Science and Research Data Management (RDM) at WSL. It supports scientists with the formal publishing of environmental datasets from forest, landscape, biodiversity, natural hazards, and snow and ice research, according to FAIR (Findability, Accessibility, Interoperability, and Reusability) principles [15,16]. The scientists' requirements are driving EnviDat's forthcoming bottom-up evolution towards integrating comprehensive visualization functionalities, including semi-automated (meta)data visualization on interactive web maps for highly heterogeneous environmental data.

Visualizing and managing large environmental geodata sets in a data publishing repository is, however, not without challenge [17]. Even basic functionalities are quite demanding, for example: (1) automated display of the environmental data contents on an interactive 2D map and/or on a 3D globe; (2) spatial, thematic and temporal navigation through a very large number of environmental data layers of a time series; and (3) extracting and displaying data subsets over an entire time series for any interactively selected location on the map. Large environmental data layers cannot be efficiently configured and interactively delivered by traditional Spatial Data Infrastructures (SDIs) and conventional Web Map Services (WMS). Implementing and managing a local SDI requires substantial computing and ample storage for supporting efficient web mapping, i.e., by creating tile caches. Furthermore, sampling, clipping, and subsetting smaller areas of interest from large time series would take a significant amount of time with any SDI's standard geoprocessing services, thereby preventing any real-time parallel interactions with a multitude of concurrent users, as required by interactive web applications. Finally, existing web-mapping frameworks are not adapted for providing user-friendly navigation along a time axis for such large numbers of data layers [17].

The challenges regarding the proper management and visualization of large environmental time series such as were uncovered during an early proof-of-concept for the implementation of web-mapping requirements in EnviDat [17]. This proof-of-concept helped evaluate some of the features that could become part of EnviDat, as well as highlighted the need for innovative concepts and technologies beyond what is presently available in contemporary state of the art [17]. Consequently, we conceptualized and implemented a new data hypercube format in order to provide user-friendly, fast, and efficient access to large time series of environmental raster data layers.

## 2. Materials and Methods

### 2.1. Concept for a New Data Hypercube Format

The data cube concept underpins the initial development of a future EnviDat web-based environmental geospatial information system (web-EGIS) module. The distinctiveness of a web-EGIS is applying geographic information systems (GIS) principles to large numbers of specialized environmental data and services, using a generic hypercube-based data organization and visualization [18]. In order to efficiently manage large time series we have conceptualized a new data hypercube-based data organization and visualization according to the following three inter-connected hypotheses:

**Hypothesis 1 (H1).** *A physical file format for raster data cubes can be defined*;

**Hypothesis 1 (H2).** *Supporting fast web visualization due to cloud-native access patterns*;

**Hypothesis 1 (H3).** *May potentially achieve more efficient compression for raster time series*.

During the conceptualization process, we attempted to take advantage of (1) the similarities between layer characteristics in such large gridded time series; (2) the use of modern encoding and decoding algorithms for compressing gridded data; and (3) the use of advanced HTML5 features available in modern web browsers. The main innovation of a hypercube data organization over a standard data cube is that multiple themes (products), locations (covering different areas of the Earth) and views (different visualizations of the same data) can be organized in a consistent manner.

First, the intrinsic characteristic of gridded data time series is that the individual data layers cover the same area of the Earth along an evenly spaced matrix/grid. Therefore, if we consider the time series a whole, all data layers have an identical spatial extent, coordinate reference system (CRS) and resolution. These similarities can be used to better store and retrieve the time series in its entirety instead of storing a series of independent image data layers. Second, changes in environmental variables and their spatial parameters can be likened to a series of successive individual frames of movie. Modern video codecs take advantage of these intrinsic characteristics to only store specific key frames and then perform motion compensation, by only encoding the differences between successive frames. They have already implemented very advanced compression algorithms for sending video data over networks without sacrificing perceived quality. Therefore, the same principles of encoding and decoding video as implemented in modern codecs can also be applied to gridded data in order to compress and transfer massive raster time series. Third, the video consumption over the web has greatly increased also due to the availability and commercial success of well-known platforms, such as Netflix or YouTube. Web browsers and HyperText Markup Language (HTML) specifications have evolved to support efficient streaming of HTML5 videos to many users in parallel. Consequently, geospatial data encoded with video codecs can be streamed efficiently to the users' web browsers as HTML5 videos. These concepts had been applied to define and implement a new data cube format named Cloud Optimized Raster Encoding (CORE) that will be gradually introduced, detailed, and explained in subsequent paper sections.

Inspired by COG, CORE was conceptualized to support state-of-the-art cloud-native access patterns but adapted to gridded time series instead of single images. The focus for this research is basic data cube format implementation, supporting fast web-based visualization and efficient access to its contained data. A proof-of-concept creation of CORE for S2 true color composite images (TCI) was implemented on a MacBook Pro (15-inch, 2017) with 16 GB RAM based on the following open-source software libraries and command line tools: GDAL [19] version "3.2.0" for raster processing and FFmpeg [20] "version N-102727-g580e168a94-tessus" for video encoding. The geodata layers from different stages of the processing workflow were visually inspected with QGIS [21] version "3.18 Zürich".

## 2.2. Methods

Fifty original S2 TCIs from an area of interest were selected from www.copernicus.eu (accessed on 15 August 2021) and transformed with GDAL for subsequent processing. The GDAL-transformed geodata layers were encoded with different quality levels using FFmpeg with H.264 encoding for generating example CORE formats for raster time series. Basic geographic and CORE-specific metadata was added using FFmpeg. Finally, the CORE-encoded S2 images were extracted back and then visually compared to the initial data in QGIS. The final CORE results as well as relevant intermediary data are fully accessible under the Document Object Identifier (DOI) mentioned in the Data Availability Statement at the end of this article. A simplified overview of the methodology is shown in Figure 1.

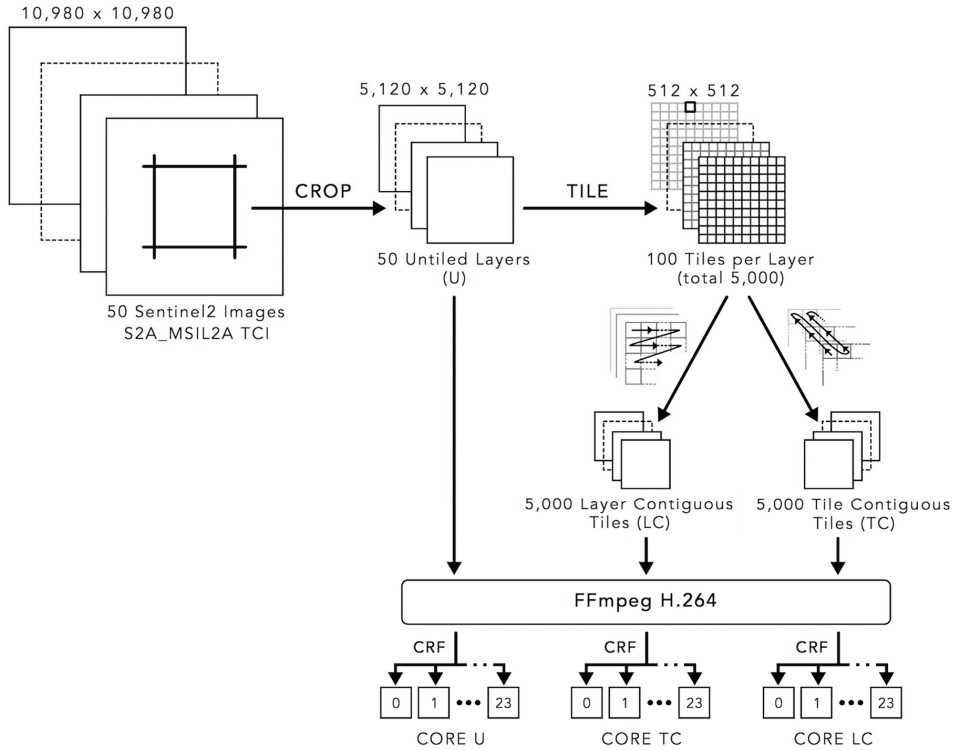

**Figure 1.** Methodology overview for creating CORE files.

### 2.2.1. Creating CORE Files for S2A Time Series

As a preparatory step, a time series of fifty (50) Sentinel 2 MSI Level-2A (S2A_MSIL2A) data were selected and downloaded via the Copernicus Open Access Hub. The original images (with dimensions of 10,980 × 10,980 pixels) are covering the same area of central–north–east Switzerland (of approximately 101 × 101 km, with a 10 m spatial resolution). The S2A_MSIL2A product packages conveniently contain TCIs in JPEG2000 format with RGB byte reflectance values (coded between 1 and 255, with 0 reserved for "no_data"). The TCI time series can be found in the "0_source" (accessible at https://envicloud.wsl.ch/#/?prefix=wsl/CORE_S2/0_source/, accessed on 10 August 2021) directory.

In the first step of the method for producing a CORE data cube, from each of the original images we have cropped smaller areas of 5120 × 5120 pixels (approximately 51 × 51 km) using GDAL as marked. These clipped areas formed our input S2A TCI time series. This clipped input series was produced because many of the source images were affected by stripes of no_data, shown as darker areas in the overlay from Figure 2, as well as by clouds. Areas containing only black (no_data) and white (clouds) compress more than cloud-free remote sensing observations. By avoiding problematic areas, we thus intentionally prevent any skewing of the compression efficiency results.

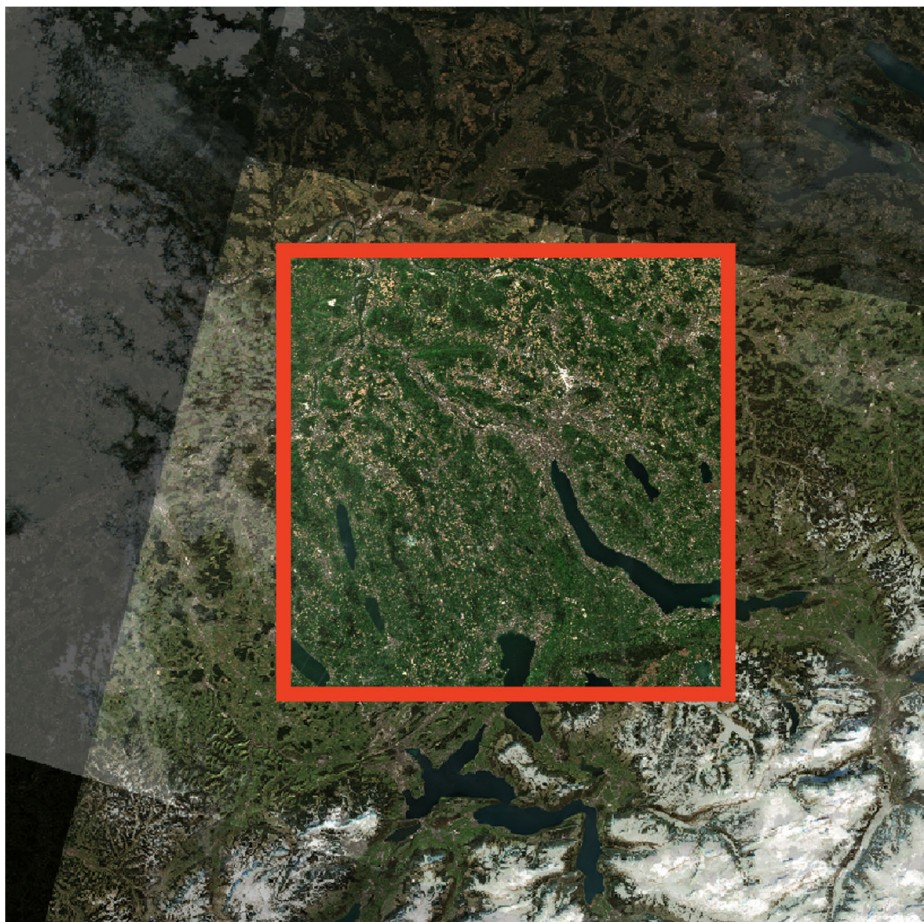

**Figure 2.** An overlay of several TCI S2A_MSIL2A images with 50% opacity, with darker shaded areas highlighting the no_data coverage stripes affecting images from the time series; the selected area is marked with the red rectangle, centered on the location of interest in Birmensdorf, CH.

Each cropped raster layer of the time series was saved as an untiled (U), full-size PNG image (with 5120 × 5120 pixels), as well as segmented in 100 smaller tiles (with each tile having 512 × 512 pixels). This resulted in a series of 50 input layers as well as 5000 tiles (50 layers × 100 tiles per layer) as illustrated in Figure 1. The 5000 tiles were ordered in two different modes: layer contiguous (LC) and tile contiguous (TC). In the LC mode, the entire time series is composed of the first 100 tiles of the first layer, followed by the next 100 tiles of the second layer, and so on. In the TC mode, the time series is composed and ordered first by all the 50 tiles covering the same region over all layers, followed by the next 50 tiles from the next tiled region over all layers, and so on. The different ordering offers faster and contiguous access to full individual layers (in LC mode) or to the same data region from all the 50 layers (in TC mode). The TC mode is especially important for fast-streamed access to a subregion of the entire time series, where tile areas with the same geographic extent from all layers are consecutively ordered one after the other. All the clipped and tiled images of time series had to be transformed in standard lossless PNGs that can be read by FFmpeg, with the georeferencing kept in external World Files (WLD) and GDAL's persistent auxiliary metadata (PAM) files. Finally, a single level of lower resolution images at 10% of the original size (512 × 512 pixels) were generated with GDAL in compressed PNG format, to serve as input for creating CORE overviews. The PNG-compressed input images of the time series from step 1 are available in directories starting with "1_input_", such as "1_input_untiled" (accessible at https://envicloud.wsl.ch/#/?prefix=wsl/CORE_S2A/1_input_untiled/, accessed on 10 August 2021).

In the second step, the PNGs were inputted to FFmpeg to encode the main CORE MP4 video format with the H.264 encoder using the x264 library [22]. The encoding of the S2 raster time series was performed from lossless PNG compressed images (for both main CORE files and the external overviews) with a Constant Rate Factor (CRF) control mode in order to achieve the highest possible quality. For the x264 encoder the "range of the CRF scale is 0–51, where 0 is lossless, 23 is the default, and 51 is worst quality possible" [22]. Consequently, we have encoded CORE files with a series of CRF values ranging from 0 (lossless) to 23 (default setting) as illustrated in Figure 1. The multiple encodings provide material for future quality analysis of the changes between CORE files at different CRF, as the CRF directly influences the achievable data compression. The frame rate was set to 1, which, in the absence of tiles, ensures that the number of S2 images in the data cube will be equal to the time duration of the video measured in seconds. When tiled, the video duration (in seconds) will be the number of layers (N) × the number of tiles per layer (M). The H.264 encoder was selected for the following key reasons: (1) wide support and compatibility in modern web browsers [23]; (2) extensive availability for hardware-assisted encoding and decoding with Intel's Quick Sync Video [24], AMD's Video Core Next [25] and NVidia's NVENC/NVDEC [26,27] and, ultimately, because of (3) the considerably faster encoding time in comparison with more recent open-source codecs, such as AOMedia Video 1 (AV1), which is currently available purely as software encoding [28].

After encoding, various metadata tags have been added also using FFmpeg, including geographic metadata tags and CORE-specific metadata tags for the number of layers, the number of tiles per layer, and the tile ordering (U, LC, TC). The main CORE files are complemented by the external overviews and WLD files. The external overviews are also mini-CORE MP4 files themselves, albeit at a lower resolution, to offer a fast first preview of the data. All created CORE files from step 2, having different CRF levels and tiling modes, are available in the "2_core_stream" (accessible at https://envicloud.wsl.ch/#/?prefix=wsl/CORE_S2A/2_core_stream/, accessed on 10 August 2021) directory, as for example: "CORE_h264_18_tc.mp4" (accessible at https://os.zhdk.cloud.switch.ch/envicloud/wsl/CORE_S2A/2_core_stream/CORE_h264_18_tc.mp4, accessed on 10 August 2021).

### 2.2.2. Extracting S2 Data Layers from CORE Files

Due to the fact that CORE is a new format, any data encoded in CORE cannot (yet) be directly read by off-the-shelf GIS software. Consequently, we provide a script to conveniently extract image layers from CORE as part of the EnviDat GitHub repository [29].

For the data extraction, the script requires the following files: (1) the CORE file (containing the data layers encoded as frames in a video container); (2) the external WLD and PAM files; and (3) an optional list of layer names corresponding to the original source data. To facilitate the reproducibility of this study, if the required files are not found locally, they will be automatically downloaded, together with a DOI link to the metadata of the dataset. The quality level of the CORE file to be exported will be requested by the script (with valid number between 0 and 23, as available remotely). The code will then proceed to extract the frames from the corresponding CORE file as georeferenced PNG files and associate them with WLD files. The WLD files are text files extending the MP4 CORE video with georeferencing information, in order to enable GDAL and other GDAL-based GIS software to correctly read and georeferenced images extracted from CORE. The CORE data frames are exported in directories starting with "3_export_CORE_h264_" followed by the quality level (e.g., 18) and the TC/LC/U tile mode (_layer_contiguous, _tile_contiguous and _untiled), as for example: "3_export_CORE_h264_18_untiled" (directly accessible at https://envicloud.wsl.ch/#/?prefix=wsl/CORE_S2A/3_export_CORE_h264_18_untiled/, accessed on 10 August 2021). The exported TCI S2 data can be subsequently used for posterior analysis and comparison of the results with the original TCI S2 input data layers.

## 3. Results

### 3.1. Cloud Optimized Data Hypercube

The essential methods described in the previous section have allowed us to successfully define and implement a CORE data cube for storing a series of fifty full-size, non-tiled, S2 TCI raster layers. All results can be reproduced using free and open-source software (FOSS) libraries.

The CORE format combines multiple layers of a time series in streamable video files. CORE can be compared to an array of COGs, with all individual COGs being compressed in one single video file. In order to introduce the CORE concept more easily, we have chosen a straightforward encoding of untiled layers. Consequently, in untiled CORE files, each layer is stored in one video frame with the frame rate of one frame per second (1 FPS) as sketched in Figure 3.

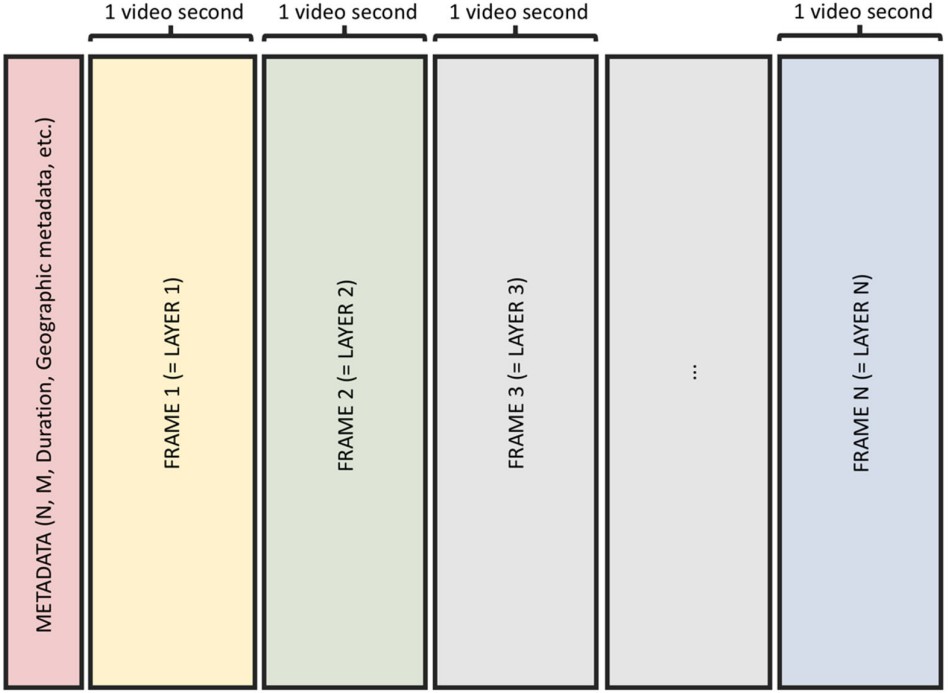

N=number of layers (50); M=number of tiles per layer (1); Duration (in seconds)=N*M=50

**Figure 3.** Schematic of a streamable CORE data cube, with metadata at the beginning of the file, followed by one data layer stored inside one frame (at one frame per second).

In production, the raster imagery should always be tiled, similarly to COG. The internal tile arrangement needs to support a fast and efficient navigation inside the CORE video format. The order of tiles as video frames in CORE is given by the main use case when accessing the data: either fast access to individual full layers or to specific subregions over all the layers in the time series. This allows for an optimized contiguous retrieval of either layers or tiles. Figure 4 sketches side-by-side the two modes of packing internal tiles in video frames: either layer contiguous (LC) or tile contiguous (TC).

Due to the 1 FPS encoding, the video duration (in seconds) represents the number of layers × the number of tiles (N*M), hence seeking and retrieving individual tiles or layers is intuitive. In our example S2A data, we have a total of 50 layers, with each layer segmented in 100 tiles. Therefore, in the S2A LC CORE example, the first 100 frames will hold all the tiles of the first layer, the next 100 the second layer and so on, with the last 100 frames of the video containing the tiles of the final layer of the time series. In the S2A TC CORE example, the first 50 frames will hold the first tile of all the 50 layers from the time series, followed by the next 50 frames containing the second tile of each 50 layers and so on, with the last 50 frames containing the last tile over the entire time series.

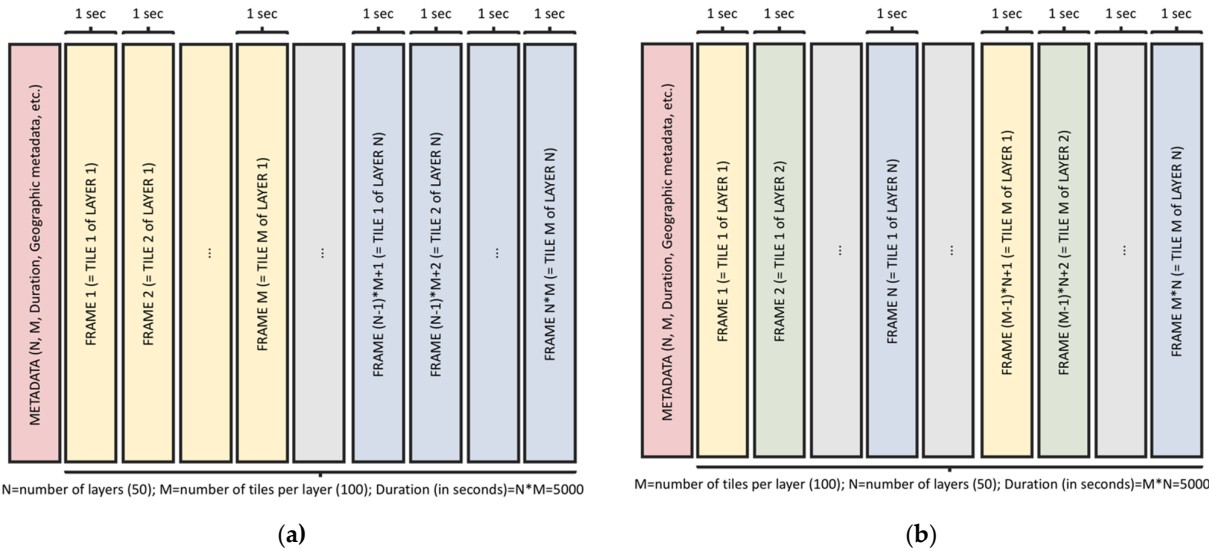

**Figure 4.** Schematics of two different internal tiling modes in CORE (**a**) layer contiguous (LC) (**b**) tile contiguous (TC).

With the above CORE results, we prove the initial Hypotheses H1 and H2, namely that a hypercube-based data organization of large raster time series is possible as a physical file (H1) for an efficient web visualization due to cloud-native access patterns (H2).

### 3.2. CORE Data Compression Gains

The third initial hypothesis is concentrated on potentially achieving a more efficient compression for raster time series (H3). Using the described reproducible methods, we have generated a series of CORE files for analyzing the compression of the entire input S2 TCI PNG time series (with 50 images). Each CORE file was encoded with a variety of CRF values starting from 0 (lossless) up to a maximum of 23 as represented in Figure 5.

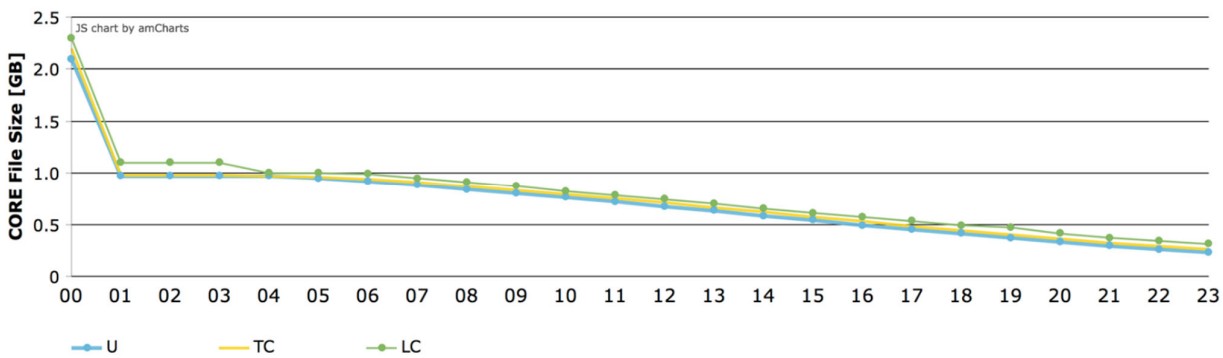

**Figure 5.** Decreasing CORE file sizes (in GB) with increasing CRF (from 0 to 23); CORE file sizes are listed in Table A1 from Appendix A, with the figure data available also online as an interactive editable chart at: https://live.amcharts.com/jA4NW/, accessed on 10 August 2021.

In the lossless mode (CRF 0), the CORE file sizes are between 2.1 GB and 2.3 GB, depending on the tile schema, which are comparable to the input data size (2.3 GB). However, as visible in Figure 5, a steady decrease of the CORE file sizes (in GB) can be observed with increasing CRF (from 0 to 23). The first significant compression gains are noted at CRF 1, which already reduces the file sizes in half when compared to the lossless versions. When reaching CRF 17–18, we can observe that the CORE files have become considerably smaller at below one quarter of the corresponding CRF 0 versions. Moreover, when reaching CRF 23, the untiled CORE file sizes has shrunk to reach almost one tenth of the sizes of the lossless versions. Furthermore, the file sizes of the LC variants are slightly larger than the TC variants, which are in turn marginally larger than the U variants.

Since the file size reduction using the H.264 video codec was unexpectedly high, we have exported the data from the corresponding CORE files in order to perform a visual data quality comparison with the original data. The capabilities of video encoding algorithms (such as H.264) to preserve the perceived data quality are exciting. A side-by-side visual comparison of selected highly zoomed regions is provided in Figure 6. CORE versions encoded with CRF 18 or less can be considered to be "visually lossless or nearly so" for the human perception [22]. Even data layers encoded with CRF 23 still retain sufficient information to visually interpret the geographical features, as visible in Figure 6d.

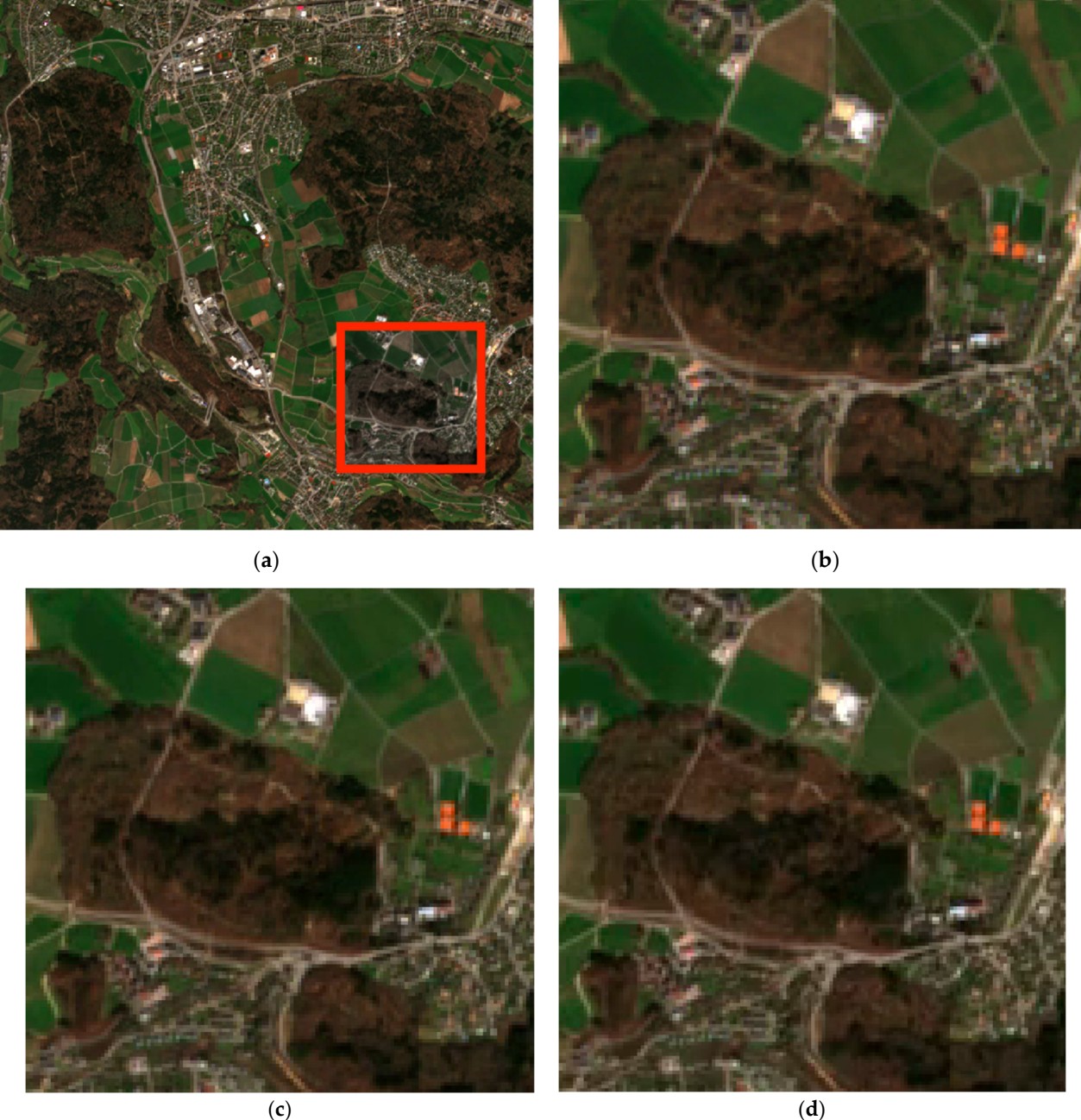

**Figure 6.** Side-by-side visual comparison between highly zoomed regions of the first image layer (layer 01, initially cropped from T32TMT_20180402T102021_TCI_10m) at different quality levels: (**a**) location of the zoomed region inside the 45th tile of layer 01. (**b**) Zoomed region from lossless CORE version encoded with CRF 0—identical with original data; (**c**) zoomed region from CORE version with CRF 18; (**d**) zoomed region from CORE version with CRF 23.

Due to our focus of the current work on the CORE format definition, an in-depth appraisal of the data quality is beyond the scope of this current work. However, it is absolutely necessary to perform a future quantitative analysis for assessing the information loss corresponding to every CORE encoding level (CRF) in its entirety. The time-consuming evaluation of the fitness of CORE-encoded imagery requires the definition of objective indicators for quantifying the degradation of data quality for a variety of typical remote sensing applications.

For the purpose of independent verification of the data quality degradation with increasing compression, we provide access to all exported data layers and tiles from the CORE files with CRF 00, CRF 01, CRF 18, and CRF 23 as georeferenced PNGs. The data is available in directories starting with "3_export_CORE_h264_" that correspond to a particular CRF and tiling mode (tile_contiguous, layer_contiguous and untiled), as for example "3_export_CORE_h264_18_untiled" (directly accessible at https://envicloud.wsl.ch/#/?prefix=wsl/CORE_S2A/3_export_CORE_h264_00_untiled/, accessed on 10 August 2021).

With the above results, we demonstrate that CORE files can compress RGB data at near lossless quality, i.e., by using a minimal lossy compression with CRF 1 in half of the lossless data size, thus supporting our initial hypothesis for the possibility of achieving a more efficient compression for raster time series (H3).

### 3.3. CORE Specifications

Based on the successful implementation of the new concept, we distillate the following CORE format specifications:

1. *A video digital multimedia container format such as MP4, WEBM, or any future video container playable as HTML video in major browsers;*

2. *Encoded using a free or open video compression codec such as H.264, AV1, or any future open-source/free video codec;*

3. *Encoding frame rate is of one frame per second (1 FPS), with each data layer segmented in internal tiles, similar to COG, ordered by the main use case when accessing the data (either LC or TC);*

4. *Optimized for streaming with the metadata placed at the beginning of the file;*

5. *Contains metadata tags for describing and using geographic raster data (similarly to the COG GeoTIFF) and CORE-specific metadata (number of layers, number of tiles, tiling schema, etc.);*

6. *Encodes similar source rasters, having the same image size and spatial resolution;*

7. *Provides lower resolution overviews for an optimized scale dependent visualization.*

The above specifications are preliminary and subject to additional improvements based on community feedback for a future standardization. To support full reproducibility, the CORE specifications and data mentioned in this study are openly published in EnviDat under a Creative Commons 4.0–CC0 "No Rights Reserved" international license, accessible at https://doi.org/10.16904/envidat.230 [30], accessed on 10 August 2021.

## 4. Discussion and Outlook

Motivated by the challenges posed by managing and visualizing large raster time series in a future web-EGIS module of EnviDat—we developed and specified the CORE format. The CORE video files were created in web-native and streamable MP4 containers, with 1 FPS. The 1 FPS is needed in order to avoid data loss performed by encoding optimizations at high frame rates, we well as to enable a straightforward seeking for individual data frames. The CORE video data is made streamable by moving metadata (moov atom) to the beginning of the MP4 file. Early access to metadata enables web browsers to access the video using cloud native access patterns with HTTP range requests without the need to download the entire file. Consequently, CORE is able to support fast data access when previewing and using geodata layers of a time series. This fast preview is enabled by the HTML5 video inherent streaming capabilities. H.264 is the fastest codec that we were able to apply in order to reliably create CORE files in a reasonable time. However, there exist newer and improved open codecs such as AV1, which is expected to achieve higher compression rates at the same perceptual quality when compared with H.264 [28].

Further research has to address the limitations of this current work. First, the handling of internal overviews needs to be developed. Currently, the overviews are external, and the goal would be to chain overviews as lower resolution videos in the same MP4 container. Internal overviews will enable fast visualization of a time series access using a single CORE file, in a similar manner to how COG works for single raster layers. Metadata encoding also needs to be improved, considering how to best encode the spatial extent, projection, layer names, and other geographical metadata. Properly defining the metadata for CORE files in future work is especially important for an appropriate implementation of the hypercube concept, where each of the layers (and by extensions its corresponding tiles) of the time series, may have a different geographical projection and/or theme. Additional research is also needed to apply CORE beyond RGB–Byte datasets. For this purpose, we will need to define extended specifications for a standardized conversion of cell values from popular GIS one-band raster data types (such as Int16, UInt16, UInt32, Int32, Float32) into multi-band red, green, blue, and alpha (RGBA) Byte values, and reconstructing them back to the original data type within acceptable thresholds. For data types that have at most 4 Bytes a standard translation to CORE is theoretically possible without information loss, because of the evolution of modern video codecs towards encoding four-channel RGBA for web videos.

In this study, fifty TCIs were selected to provide a realistic example of a data cube for time series, while keeping the processing time of subsequent steps at reasonable levels and enable reproducibility on standard computing hardware. Future work will test the applicability of the data hypercubes to the management of many thousands of original high-resolution RGB tiles for different products, with the note that a High-Performance Computing Cluster (HPC) will be needed for processing and encoding such a massive amount of imagery into CORE.

Finally, the evaluation of the fitness of CORE-encoded imagery for remote sensing applications is essential for future research and development of the CORE format. Lossless data compression with CRF 0 will be able to preserve the data values but it will forego significant compression gains. A slight lossy encoding with CRF 1 is able to cut the data size in half, while compressing with higher CRF will reduce the CORE files to a fraction of the input. Consequently, we can obtain significant storage savings and faster network data transmission, but it is unknown how much lossy encoding can be tolerated by various scientific applications. We need objective indicators for assessing the information loss corresponding to different CORE CRF encoding levels, in order to accurately understand how compression-caused value errors will influence data or model uncertainty, especially for applications where error inflation in subsequent analysis is of concern.

## 5. Conclusions

User requirements related to mapping and visualization are transforming EnviDat into a next-generation environmental data-publishing portal that fuses publication repository functionalities with web-EGIS and EO data cube functionalities. The key CORE innovation is reusing and adapting COG principles to a collection of many layers in a time series. Time series of EO data or climate model outputs are especially suited for an efficient encoding and publication in CORE, instead of complex directory structures containing large numbers of individual files. The CORE format enables an efficient storage and management of uniform gridded data by applying video encoding algorithms. The layers should be segmented in smaller internal tiles for enabling a faster access to the data, in a similar manner to how COG works for single raster layers. The tiling schema should be adapted to the main data use case, by using a specific ordering of the tiles following a TC/LC approach. The CORE format has native support for data hypercubes. CORE can contain any number of raster data layers with uniform pixel size and image size dimensions, irrespective of their theme or spatial extent. Another innovation is that CORE, unlike COG, specifically uses a web format container that can be decoded by major web browsers, hence enabling a web-EGIS to provide user-friendly, fast, and efficient access to CORE data hypercubes, containing considerable numbers of raster data layers in a time series. The design of CORE is based on the COG principles for enabling more efficient workflows on the cloud, but adapted to solving web-EGIS visualization challenges for large environmental time series in geosciences.

The results obtained so far demonstrate a possible solution for an effective management and visualization of large environmental raster time series by significantly reducing their overall size at near lossless quality. Consequently, CORE is suitable not only for a faster web visualization but also for a more efficient exchange and preservation of time series raster data in environmental data repositories such as EnviDat. Geospatial information has the potential to become a central element for any specialized environmental data portal. With the development of time series in CORE format, EnviDat will further improve its capabilities for presentation, documentation, and exchange of scientific geoinformation. The CORE format specifications are open source and can be used by other platforms to manage and visualize large environmental time series. We hope EnviDat can serve as a model for other FAIR platforms and repositories that are being specifically developed for the management and publication of environmental data.

**Author Contributions:** Conceptualization, I.I.E., D.H.-A., L.d.E., M.R., R.K.B., D.H., D.N.K., G.-K.P., M.H., C.G., N.E.Z. and L.P.; methodology, I.I.E., L.d.E.; software, I.I.E., D.H.-A., L.d.E., D.H. and R.K.B.; validation, I.I.E., D.H.-A., L.d.E., D.H., M.R. and R.K.B.; formal analysis, I.I.E.; investigation, I.I.E., D.H.-A., L.d.E., D.H., M.R., D.N.K. and R.K.B.; resources, G.-K.P. and M.H.; data curation, I.I.E. and R.K.B.; writing—original draft preparation, I.I.E.; writing—review and editing, I.I.E., L.d.E., M.R., R.K.B., D.H., D.N.K., M.H., C.G., N.E.Z. and L.P.; visualization, I.I.E., L.d.E.; supervision, I.I.E.; project administration, G.-K.P. and I.I.E.; funding acquisition, G.-K.P. All authors have read and agreed to the published version of the manuscript.

**Funding:** This research received no external funding.

**Data Availability Statement:** The data presented in this study are openly available in EnviDat, at https://doi.org/10.16904/envidat.230 accessed on 15 August 2021 Please note that to stream large untiled CORE files you will require Google Chrome and a computer that supports 8K UHD video playback.

**Acknowledgments:** We thank the members of the GIS group at WSL, the members of the map@wsl interest group and the members of the EnviDat user group for their valuable requirements, feedback, and support towards the development of GIS capabilities in EnviDat.

**Conflicts of Interest:** The authors declare no conflict of interest.

## Appendix A

Table A1 reviews the file sizes obtained for the different CORE files with a specific CRF encoding, as uniformly listed in the cloud at https://envicloud.wsl.ch/#/?prefix=wsl/CORE_S2A/2_core_stream/, accessed on 10 August 2021.

**Table A1.** Variation of the file size with increasing CRF.

| CORE Quality [CRF] | File Names | U [1] File Size [GB] | TC [2] File Size [GB] | LC [3] File Size [GB] |
|---|---|---|---|---|
| 0 | CORE_h264_00(_*).mp4 | 2.1 | 2.2 | 2.3 |
| 1 | CORE_h264_01(_*).mp4 | 0.97 | 0.98 | 1.1 |
| 2 | CORE_h264_02(_*).mp4 | 0.97 | 0.98 | 1.1 |
| 3 | CORE_h264_03(_*).mp4 | 0.97 | 0.98 | 1.1 |
| 4 | CORE_h264_04(_*).mp4 | 0.97 | 0.97 | 1.0 |
| 5 | CORE_h264_05(_*).mp4 | 0.95 | 0.96 | 1.0 |
| 6 | CORE_h264_06(_*).mp4 | 0.92 | 0.94 | 0.99 |
| 7 | CORE_h264_07(_*).mp4 | 0.88 | 0.91 | 0.95 |
| 8 | CORE_h264_08(_*).mp4 | 0.84 | 0.87 | 0.91 |
| 9 | CORE_h264_09(_*).mp4 | 0.80 | 0.83 | 0.87 |
| 10 | CORE_h264_10(_*).mp4 | 0.76 | 0.79 | 0.82 |
| 11 | CORE_h264_11(_*).mp4 | 0.72 | 0.75 | 0.78 |
| 12 | CORE_h264_12(_*).mp4 | 0.67 | 0.71 | 0.74 |
| 13 | CORE_h264_13(_*).mp4 | 0.63 | 0.66 | 0.70 |
| 14 | CORE_h264_14(_*).mp4 | 0.58 | 0.62 | 0.65 |
| 15 | CORE_h264_15(_*).mp4 | 0.54 | 0.57 | 0.61 |
| 16 | CORE_h264_16(_*).mp4 | 0.49 | 0.53 | 0.57 |
| 17 | CORE_h264_17(_*).mp4 | 0.45 | 0.48 | 0.53 |
| 18 | CORE_h264_18(_*).mp4 | 0.41 | 0.44 | 0.49 |
| 19 | CORE_h264_19(_*).mp4 | 0.37 | 0.40 | 0.47 |
| 20 | CORE_h264_20(_*).mp4 | 0.33 | 0.36 | 0.41 |
| 21 | CORE_h264_21(_*).mp4 | 0.29 | 0.32 | 0.37 |
| 22 | CORE_h264_22(_*).mp4 | 0.26 | 0.29 | 0.34 |
| 23 | CORE_h264_23(_*).mp4 | 0.23 | 0.26 | 0.31 |

[1] U—contains untiled, full frame size raster data; [2] TC—tile_contiguous raster data; [3] LC—layer_contiguous raster data.

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
