# Peer review of "Cloud Optimized Raster Encoding (CORE): A Web-Native Streamable Format for Large Environmental Time Series"

_2673-7418, doi:10.3390/geomatics1030021_

Round 1
Reviewer 1 Report
The paper "Cloud Optimized Raster Encoding (CORE): A Web-Native Streamable Format for Large Environmental Time Series" aims on developing a new web format for raster data. An article is based on new cloud
optimized raster encoding format which is an extremly useful topic within geospatial world.
I would like to congratulate authors to an excelent paper. The paper is well structured, it follows an academic structure, it consist of generapl concept and practical implementation which provide quite interesting outputs to the readers.
The workflow is inspired (as authors mentioned in the paper)by the cloud optimized GeoTiff but the article introduces brand new implementation. The combination of particular methods provides extremly interesting results which could be easily transferable. Images/maps are well prepared and designed.
I have just a small notice. If the concept follows open source concept, I miss furter information - where the documentation is available? Under which license will be the format published?
Therefore I do not have any significant comments, and I recommend to accept the paper in the present form for publishing in Geomatics journal.
Author Response
Thank you very much for your positive review.
As documented in the data accessibility section, the specifications were released publicly under CC0 license (no rights reserved) at https://www.doi.org/10.16904/envidat.230
to quote: „The Cloud Optimized Raster Encoding (CORE) specifications are released to the public domain under a Creative Commons 4.0 CC0 "No Rights Reserved" international license. You can reuse the information contained herein in any way you want, for any purposes and without restrictions.“
There are no strings attached - the implementation in software is moving slowly due to our small development team, so we would very much welcome any external implementations of the CORE specifications that can be done faster by other research teams.
Reviewer 2 Report
see comments in the paper
Data storage and visualisation is only one part, the web-based processing (e.g. index computation) seems to be of more interest for users
Rasdaman works with imagery time series? Comments on their data model and functionality compared to yours

Author Response
Dear reviewer,
Many thanks for your valuable comments, they were addressed as follows:
line 50 – modified the paragraph to be explicit instead of implied:
"The data volume produced by the Sentinel missions is quite significant, with one single compressed (.zip) tile having around 1 GB. This is due to the fact that the Multi Spectral Instrument (MSI) of the Sentinel-2 (S2) mission produces high spatial resolution data (from 10 m to 60 m depending on the spectral band) with a total of 13 bands covering the visible to the short-wave infrared [12]."
line 65 – yes, FAIR principles as referenced from ref. no. 16 (https://doi.org/10.5194/egusphere-egu21-5663); the paragraph was modified accordingly:
"EnviDat actively promotes good practices for Open Science and Research Data Management (RDM) at WSL. It supports scientists with the formal publishing of environmental datasets from forest, landscape, biodiversity, natural hazards and snow and ice research according to FAIR (Findability, Accessibility, Interoperability and Reusability) principles [15,16]."
line 104 – corrected, thank you for spotting the typo
lines 119/120 – the current study is limited to identical CRS, and therefore the text has been modified accordingly: "all data layers have an identical spatial extent, coordinate reference system (CRS) and resolution"
However, FYI, this doesn’t imply that the CORE format is not able to be used with different CRS. Quite the opposite! The CRS and georeferencing information are metadata, while only the image size (in pixels) and the ground resolution of each pixel are relevant for data encoding. Consequently, if the image size and pixel size are identical between layers, CORE is definitely applicable and useful to such a multi-CRS data cube use case.
line 138 – CORE being a new format, may seem very alien in the beginning. Consequently, we have chosen (on purpose), to gradually (step by step) introduce the concept over several sections, which increasing information levels because there are a lot to explain and we don’t want to overload the reader in one single section. As you may have noticed, the format is explained with an increasing amount of detail and precision in sections 2.3, 3.1 and 3.3, with the sections 2.3.2 and 3.2 serving as relevant breaks for the reader, to recover from the main information flood…
In the commented paragraph we just introduce the concept and we make reference to COG to serve as an introduction, but we can signal that the required information is available in the later sections:
"These concepts had been applied to define and implement a new data (hyper)cube format named Cloud Optimized Raster Encoding (CORE) that will be gradually introduced, detailed and explained in subsequent paper sections."
line 194 – CRS is metadata and therefore different from image data.
line 349 – Many thanks for the suggestion and for your confidence in CORE. I have modified the paragraph as follows:
"The above specifications are preliminary and subject to additional improvements based on community feedback for a future standardization. "
Regarding the comment from the abstract about processing (and by extension about rasdaman): the CORE format is specifically designed for storage, mapping and visualization and not processing. Processing like reclassification implies a computational component, which is outside of the scope of any format. Rasdaman can support processing queries because it is not a format but an Array DBMS. When comparing CORE with Rasdaman is like comparing COG/GeoTIFF with ST_Raster from ProgreSQL/PostGIS. PostgreSQL is also a DBMS - a Database Management System - and contains a lot of application logic to support querying and processing functionalities. A format like GeoTIFF and CORE, is independent of any specific software/code. To summarize, CORE is a data format for time series (like COG), and therefore not directly comparable with a DBMS like Rasdaman.
Man thanks again for your review and hope that we could address your comments in a satisfactory manner.
Best regards,
the authors